# Degeneracy in the neurological model of auditory speech repetition

Noor Sajid [1✉], Andrea Gajardo-Vidal[1,2], Justyna O. Ekert[1], Diego L. Lorca-Puls[1,3], Thomas M. H. Hope [1], David W. Green[4], Karl J. Friston [1] & Cathy J. Price[1]

Both classic and contemporary models of auditory word repetition involve at least four left hemisphere regions: primary auditory cortex for processing sounds; pSTS (within Wernicke's area) for processing auditory images of speech; pOp (within Broca's area) for processing motor images of speech; and primary motor cortex for overt speech articulation. Previous functional-MRI (fMRI) studies confirm that auditory repetition activates these regions, in addition to many others. Crucially, however, contemporary models do not specify how regions interact and drive each other during auditory repetition. Here, we used dynamic causal modelling, to test the functional interplay among the four core brain regions during single auditory word and pseudoword repetition. Our analysis is grounded in the principle of degeneracy—i.e., many-to-one structure-function relationships—where multiple neural pathways can execute the same function. Contrary to expectation, we found that, for both word and pseudoword repetition, (i) the effective connectivity between pSTS and pOp was predominantly bidirectional and inhibitory; (ii) activity in the motor cortex could be driven by either pSTS or pOp; and (iii) the latter varied both within and between individuals. These results suggest that different neural pathways can support auditory speech repetition. This degeneracy may explain resilience to functional loss after brain damage.

[1] Wellcome Centre for Human Neuroimaging, QS Institute of Neurology, University College London, London, UK. [2] Centro de Investigación en Complejidad Social (CICS), Universidad del Desarrollo, Concepción, Chile. [3] Sección de Neurología, Departamento de Especialidades, Facultad de Medicina, Universidad de Concepción, Concepción, Chile. [4] Experimental Psychology, University College London, London, UK. ✉email: noor.sajid.18@ucl.ac.uk

Auditory speech repetition involves the immediate repro-
duction of heard speech. It requires the successful trans-
lation of auditory input into a motor output that matches
the heard speech. The influential 19th century neurological model
of language[1–4], later refined by Norman Geschwind[4], posited that
speech repetition involves a sequential flow of information across
four left hemisphere brain regions: the primary auditory cortex,
the left posterior superior temporal cortex (Wernicke's area), the
left posterior inferior frontal gyrus (Broca's area) and the primary
motor cortex, with information being relayed from Wernicke's to
Broca's areas via the arcuate fasciculus. More recent studies have
challenged this model of the functional anatomy of language. For
example, the brain regions activated during speech repetition
include multiple cortical and subcortical areas that are not part of
the neurological model[5]. Likewise, white matter tracts, other than
the arcuate fasciculus, have been shown to connect temporal and
frontal regions[6,7]. These and other observations led Tremblay and
Dick[8] to claim that the Wernicke-Lichtheim-Geschwin model is
obsolete. More contemporary models of auditory speech repeti-
tion recognise the existence of direct and indirect connections
between parts of Wernicke's area and Broca's area, with the
indirect pathway routed via the temporo-parietal junction (also
referred to as the sylvian parietal temporal area, Spt) and/or the
supramarginal gyrus[9,10]. However, despite widespread doc-
umentation that the neurological model of language is over-
simplified, there is strong functional neuroimaging evidence
showing that left temporal and frontal regions in the vicinity of
Wernicke's and Broca's areas contribute to auditory speech
repetition. Thus, one can argue that the classic neurological
model captures the core of the auditory speech repetition system,
while acknowledging that other cortical and subcortical regions
are also involved.

The current paper investigates how Wernicke's and Broca's
areas interact with one another during auditory speech repetition,
and how they are driven by, and/or drive responses in left
auditory cortex (A1), and the left primary motor cortex (M1).
Our prior fMRI studies of auditory word and pseudoword repe-
tition[5] identified activation in: dorsal and ventral parts of the pars
opercularis (dpOp and vpOp) in the vicinity of Broca's area, the
rostro-posterior superior temporal sulcus (pSTS), in the vicinity
of Wernicke's area, primary auditory cortex (A1) and motor
cortex (M1) including both the face region (M1-f) and the tongue
and larynx region (M1-tl) in addition to other areas that are not
under investigation here. To ensure accurate anatomical locali-
sation, regions of interest were defined using the Brainnetome
atlas[11] (Fig. 1a), and the time series of activation during auditory
word and pseudoword repetition was extracted from the peak
voxel within each anatomically defined region of interest for 59
neurotypical participants. Considering the two subdivisions of
pOp (dpOp and vpOp)—and of M1 (M1-f and M1-tl)—there
were four possible regional configurations that include A1 and
pSTS with either: (1) dpOp and M1-f, (2) vpOp and M1-f,
(3) dpOp and M1-tl, and (4) vpOp and M1-tl.

Directed interactions (i.e., effective connectivity) across activity
in A1, pSTS, pOp and M1, during auditory repetition, was
assessed using Dynamic Causal Modelling (DCM)[12–15]. Contrary
to the classic neurological model, we did not assume a serial
relationship among the four regions but tested for bilateral con-
nections between all regions, except the input (A1) and output
(M1) regions. Furthermore, our analyses were agnostic as to the
white matter tracts mediating effective connectivity. For example,
effective connectivity from pSTS to pOp would not necessarily
imply monosynaptic connections via a direct white matter tract
(e.g., the arcuate fasciculus). It could also be the consequence of
indirect (polysynaptic) connections (e.g., via Spt or SMG) or even
multiple fasciculi. In other words, effective connectivity can be
mediated vicariously through intervening cortical stations (not
included in the model).

If the classic and contemporary models of auditory word
repetition are correct, our DCM analysis should show that activity
in the primary auditory cortex excites pSTS, pSTS excites pOp
and pOp excites the primary motor cortex. Evidence for this
model was compared against evidence for alternative models that
allowed, for example, (a) primary motor cortex to be driven by
pSTS as well as pOp instead of pOp only and (b) inhibitory as
well as excitatory extrinsic (i.e., between-region) connectivity. We
were also interested in whether effective connectivity varied with
task (word or pseudoword repetition) and whether there was
evidence for degeneracy in functional architectures; namely,
whether effective connectivity varied across participants (inter-
subject variability) or within participants (intra-subject varia-
bility) for the same task. Evidence for degeneracy in terms of
variation within and between participants, would provide insights
into how language functions recover following neurological
damage. Under this formulation, structures (e.g., subgraphs of a
neuronal network) may be sufficient, but not necessary, for a
particular function—meaning that functional deficits arise only
when all degenerate neural pathways are damaged[16–18].

## Results

Effective connectivity was assessed with Bayesian model com-
parison – which compares the evidence for models with and
without each connection. The strength of connection is quantified
in terms of posterior estimates of the model's connectivity
parameters. For extrinsic (i.e., between region) connections, a
positive connection is excitatory (i.e., activity in one region
increases activity in another), while a negative connection is
inhibitory (i.e., activity in one region decreases activity in
another). For intrinsic (within region) connections, positivity
means that increased activity results in greater self-inhibition,
whereas negativity means increased activity results in less self-
inhibition. Throughout the results, tables, and figures, we only
report estimated connectivity with a posterior probability greater
than 0.75.

**Group-level effective connectivity for word and pseudoword
repetition**. For word repetition, we observed excitatory extrinsic
connectivity from A1 to pSTS, pSTS to M1, A1 to pOp, and pOp
to M1. In addition, there were inhibitory effective connections
from M1 to A1, pSTS to A1, and most surprisingly, between pSTS
and pOp in both directions. The same results were observed for
different subregional configurations: i.e., replacing the dorsal pOp
(dpOp) subregion with the ventral pOp (vpOp) subregion and/or
the face motor control (M1-f) subregion with the tongue and
larynx motor control (M1-tl) subregion (Fig. 1b and Table 1). The
main difference between subregions was that self-inhibition for
pSTS was greater when M1-f was included compared to when
M1-tl was included.

For pseudoword repetition, the estimated group-level con-
nectivity was very similar to that observed for word repetition
(Fig. 1c and Table 1). In particular, during both auditory
pseudoword and word repetition, there was evidence for (i) a
connection from pSTS to M1 that was independent of pOp and
(ii) an inhibitory connection from pOp (both dorsal and ventral)
to pSTS in all four configurations and (iii) an inhibitory
connection from pSTS to pOp for all configurations except dpOp
and M-f.

In contrast to word repetition, effective connectivity for
pseudoword repetition identified: (i) no inhibitory (or excitatory)
connection from pSTS to dpOp in the configuration with M1-f;
(ii) a negative (i.e., less inhibitory) self-connection for pSTS (iii) a

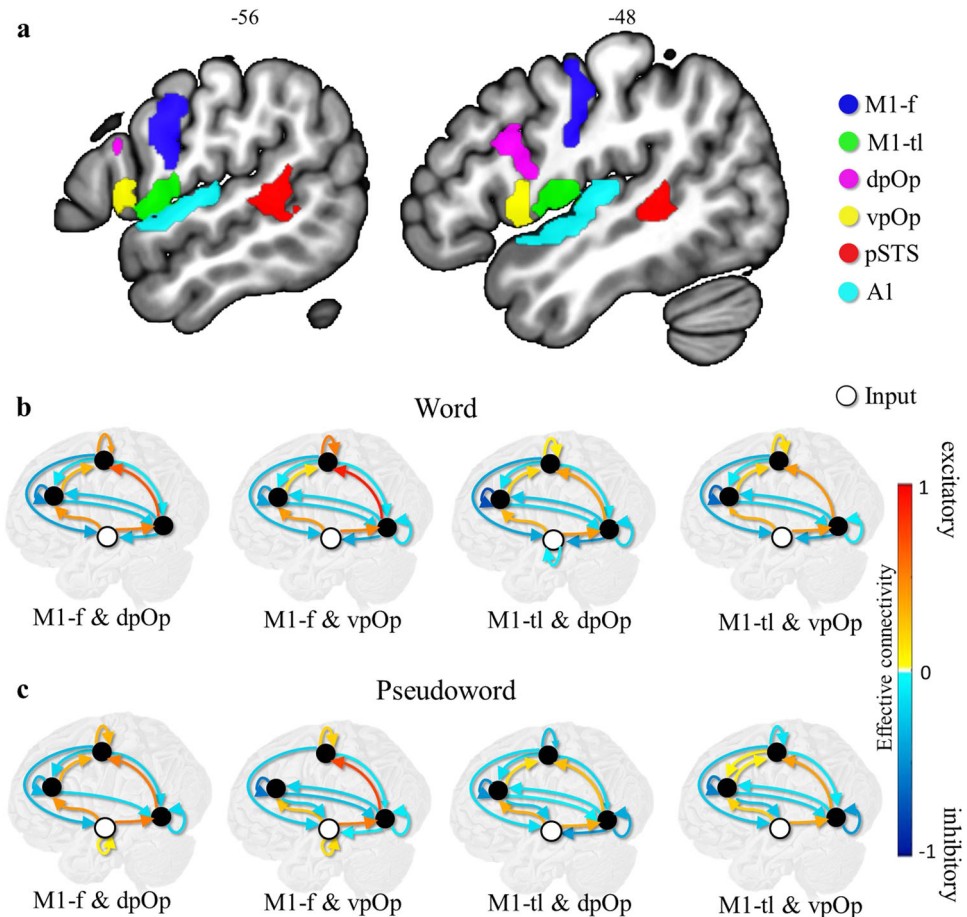

**Fig. 1 Estimated effective connectivity for the anatomical regions of interest. a** shows the 6 anatomical regions of interest: M1-f (blue), M1-tl (green), dpOp (magenta), vpOp (yellow), pSTS (red) and A1 (cyan). **b** shows the strength of effective connectivity among regions, during word repetition, after Bayesian model selection. In each model, the four circles represent A1 (bottom, white circle, input area), pSTS (black circle, right), pOp (black circle, left) and M1 (black circle, top). Four different models are depicted for either vpOp or dpOp; with either M1-f or M-tl. **c** shows the same for pseudoword repetition. Red lines denote positive (excitatory) extrinsic connections between regions, with a maximum value of 1, dark blue denotes negative (inhibitory) extrinsic connections between regions, with a maximum value of −1. The other lines represent connections graded within these extremes, see Table 1 for details. Self-connections with a high posterior probability (i.e., >0.75 representative of strong Bayesian evidence) are log scale parameters that scale inhibitory intrinsic connectivity.

**Table 1 Estimated connections at the group level, in each model configuration, for word and pseudoword repetition.**

| Model | | Word repetition | | | | Pseudoword repetition | | | |
|---|---|---|---|---|---|---|---|---|---|
| | | M1-f & dpOp | M1-f & vpOp | M1-tl & dpOp | M1-tl & vpOp | M1-f & dpOp | M1-f & vpOp | M1-tl & dpOp | M1-tl & vpOp |
| **From** | **To** | | | | | | | | |
| A1 | A1 | (0.00) | (0.00) | −0.09 | (0.00) | 0.07 | 0.15 | (0.00) | (0.00) |
| A1 | pSTS | 0.51 | 0.49 | 0.42 | 0.42 | 0.53 | 0.52 | 0.33 | 0.35 |
| A1 | pOp | 0.42 | 0.39 | 0.31 | 0.32 | 0.43 | 0.24 | 0.33 | 0.20 |
| pSTS | A1 | −0.28 | −0.36 | −0.44 | −0.35 | (0.00) | −0.09 | −0.15 | (0.00) |
| pSTS | pSTS | (−0.06) | −0.15 | −0.18 | −0.10 | −0.18 | −0.10 | −0.37 | −0.43 |
| pSTS | pOp | −0.27 | −0.18 | −0.25 | −0.20 | (−0.06) | −0.20 | −0.14 | −0.13 |
| pSTS | M1 | 0.66 | 0.88 | 0.38 | 0.41 | 0.53 | 0.73 | 0.27 | 0.38 |
| pOp | A1 | (0.00) | (−0.05) | (0.00) | (−0.05) | (0.00) | −0.19 | −0.25 | −0.26 |
| pOp | pSTS | −0.22 | −0.21 | −0.16 | −0.18 | −0.17 | −0.27 | −0.08 | −0.18 |
| pOp | pOp | −0.49 | −0.40 | −0.76 | −0.63 | −0.32 | −0.58 | −0.48 | −0.54 |
| pOp | M1 | 0.31 | 0.13 | 0.18 | 0.19 | 0.32 | (0.00) | 0.15 | 0.06 |
| M1 | A1 | −0.40 | −0.29 | −0.38 | −0.33 | −0.30 | −0.20 | −0.14 | −0.25 |
| M1 | pSTS | −0.11 | −0.12 | −0.13 | −0.04 | −0.19 | −0.10 | −0.16 | −0.09 |
| M1 | pOp | −0.13 | −0.10 | −0.13 | −0.10 | −0.21 | (0.00) | −0.15 | 0.05 |
| M1 | M1 | 0.41 | 0.52 | 0.09 | 0.15 | 0.31 | 0.20 | −0.26 | −0.13 |

Extrinsic (between-region) connections are parameterised directly as rate constants in units of hertz (Hz) because they are rates of change (i.e., the rate of change in a target region, per unit change in the source region). In contrast, intrinsic (self) connections in DCM for fMRI are log-scaling parameters that are applied to inhibitory connections to ensure dynamical stability. This means that a positive self-connection means greater self-inhibition. Connections with a posterior probability less than 0.75 are shown, for completeness in brackets.

negative (i.e., less inhibitory) self-connection for M1-tl, and (iv) no inhibitory connection from pSTS to A1.

Speculatively, these task-specific effects may have arisen because, under a predictive coding account, there may be lower precision in predictions for auditory processing when the stimuli are always unfamiliar within a run (i.e., the pseudoword condition) compared to when the stimuli are always familiar within a run (i.e., the word condition). We do not discuss these results further.

**Variation in connectivity from pOp and pSTS to M1, across participants and models**. To investigate the (unanticipated) group-level effective connectivity from pSTS to M1 in addition to pOp to M1 (Fig. 1), we evaluated how effective connectivity from pOp and pSTS to M1 varied across participants and models (Fig. 2a, b; Supplementary Figure 1, Table 1 and Table 2).

For each participant, we considered 8 combinations of subregions per task; interchanging the two M1 regions (M1-f or M1-tl) with the two pOp regions (dpOp or vpOp) for each auditory speech repetition task (word or pseudoword). Each model was assigned to one of four groups (A–D in Fig. 2a, b), according to the presence or absence of significant excitatory connectivity from pOp or pSTS to M1 (posterior probability >0.75). Group A was defined by excitatory connections from both pOp and pSTS to M1; Group B was defined by excitatory connections from pSTS to M1 but not from pOp to M1; Group C was defined by excitatory connections from pOp to M1 but not from pSTS to M1; and Group D was defined by the absence of definitive connections from both pOp and pSTS to M1 (i.e., a posterior probability range of <0.75).

Across participants and subregional configurations, more than half the models included excitatory connections from both pOp and pSTS to M1 (i.e., Group A; Fig. 3a, c, d). Those with excitatory connections from pSTS to M1 but not from pOp to M1 (Group B) and those without connections from either pOp or pSTS to M1 (Group D) accounted for a further ~20% of estimated models. Importantly, <5% of estimated models fell in Group C (i.e., excitatory connections from pOp to M1 but not from pSTS to M1). Moreover, only 2% of models (10 in total) were consistent with the neurological model, i.e., Group C with excitatory connections from pSTS to pOp (pSTS->pOp->M1), see Fig. 3b–d.

No significant differences in the proportion of A, B, C or D models were observed for (i) word versus pseudoword repetition

(Fig. 3a) and (ii) subregional configurations including activity from vpOp versus dpOp or from face (M1-f) versus the larynx and tongue (M1-tl) (Supplementary Fig. 2) irrespective of whether or not p values were corrected for multiple comparisons (i.e., Mann-Whitney-Wilcoxon two-sided tests with or without Bonferroni correction). In addition, we did not detect group differences in (iii) the degree of activation in any of the subregions (Supplementary Fig. 3) or (iv) behavioural performance across different subregional configurations (Supplementary Tables 1, 2).

**Degeneracy in auditory repetition**. Degeneracy pertains to how many independent structures or neural pathways can be recruited to subserve the same function or task[15,16,19], for example, auditory repetition of heard speech. In our results, degeneracy in the neural pathways supporting auditory speech repetition is implied by the high inter- and intra-subject variability in group membership, i.e., variability in the degree to which M1 activity was driven by pSTS, pOp, both or neither. Inter-participant variability in connectivity, within the same task and within the same regional configuration, indicates that there are different ways that the same task can be performed across participants. Intra-participant variability in connectivity was only assessed across tasks and regional configurations. In other words, we assessed intra-subject variability as the number of connectivity groups (1-4) that each participant was assigned to. For example, during word repetition, subject C073 belonged to both Group B (3x) and Group C (1x). Details for all participants are shown in Supplementary Data 1.

Intra-participant variability in connectivity was only assessed across tasks and regional configurations. It illustrates that participants have the capacity to use different neural pathways. Theoretically, such variation could be the consequence of task or regional configuration. However, this is unlikely because (i) inter-patient variability was observed when task and regional configuration was controlled and (ii) there were no consistent differences in which connections were used for words and pseudowords (see Fig. 3). In information theory, the amount of uncertainty or randomness in a system can be quantified as entropy and measured in units of nats, which is a logarithmic scale based on the natural logarithm with a base of $e$. This is particularly useful when dealing with probabilities that are not evenly distributed and has previously been used to denote

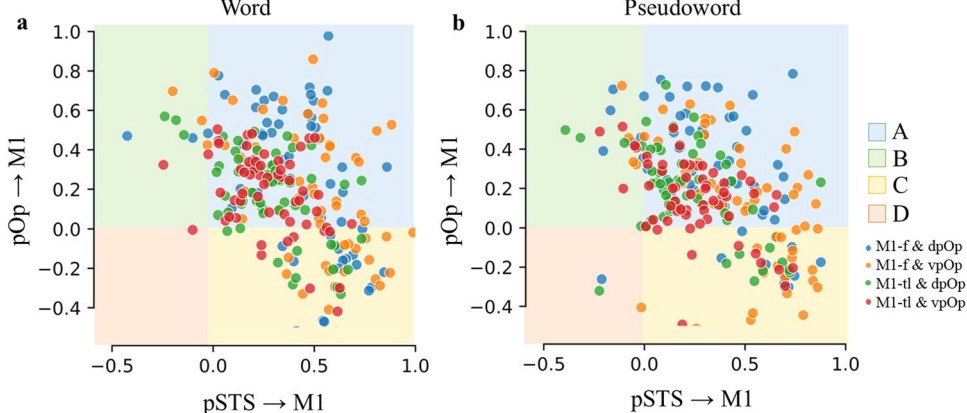

**Fig. 2 Individual-level effective connectivity from pOp and pSTS to M1.** For word (**a**) and pseudoword (**b**) repetition separately, group membership (A, B, C or D) is illustrated for each of the four subregional configurations for each participant. The colour dots denote the model specification, i.e., blue is for the M1-f and dpOp, orange for M1-f and vpOp, green for M1-tl and dpOp and red for M1-tl and vpOp. Group A included excitatory connections from both pOp and pSTS to M1; Group B included excitatory connections from pSTS to M1 but not from pOp to M1; Group C included excitatory connections from pOp to M1 but not from pSTS to M1; and Group D did not include excitatory connections from either pOp or pSTS to M1.

degeneracy[19]. In our results, low entropy (=0) denotes membership of a single group (e.g., the participant was consistently in group A) whereas high entropy (>1.3) denotes membership dispersed over all groups (i.e., the participant was in group A, B, C and D for the different subregional combinations or tasks). We found that the average entropy in individual group membership was 0.49 for word repetition and 0.55 for pseudoword repetition (Fig. 4) and the majority (73%) of our participants were assigned to at least 2 different groups, indicating that they could execute the auditory speech repetition tasks in different ways (i.e., degeneracy).

Importantly, this intra-participant variability, and the similarity of the models we observe at the group level for words and pseudowords, makes it highly unlikely that inter-participant variability can be explained solely in terms of between-participant variability in brain structure and functional anatomy.

## Discussion

The classic neurological model of language[1–4] posits that, during auditory speech repetition, information flows sequentially from A1 to Wernicke's area, Wernicke's area to Broca's area (via the arcuate fasciculus), and finally Broca's area to M1. Contemporary models of auditory speech repetition incorporate greater anatomical precision (e.g., here we use pSTS as a proxy for Wernicke's area and pOp as a proxy for Broca's area) and posit direct and indirect connections between these temporal and frontal areas, due to the involvement of other regions (e.g., in temporo-parietal cortex and subcortical structures). However, the general consensus across classic and contemporary models is that pSTS connects directly or indirectly with M1 through pOp (and other regions)[9,20].

In this study, we used DCM of fMRI data from neurotypical participants to evaluate the effective connectivity between pSTS and pOp, and between each of these regions and A1 and M1. In agreement with prior models, we found that, during word and pseudoword repetition, there were excitatory forward connections from pOp to M1 at the group level. However, in contradiction with prior models: (i) pSTS exerts an excitatory influence on M1 that cannot be explained by afferents from pOp; (ii) the connections between pSTS and pOp were inhibitory rather than excitatory, in both directions; (iii) there were profound inter-

**Table 2 Summary of participant.**

| Participants | |
| --- | --- |
| Sample size | 59 |
| Gender (female; male) | 34;25 |
| Age in years (±std.) | 44.5 (17.66) |
| Word repetition | |
| Reaction time in msec (±std.) | 1163.28 (158.13) |
| Accuracy as % correct (±std.) | 99.48 (1.38) |
| Pseudoword repetition | |
| Reaction time in msec ±std.) | 1249.65 (204.81) |
| Accuracy as % correct (±std.) | 97.80 (4.41) |

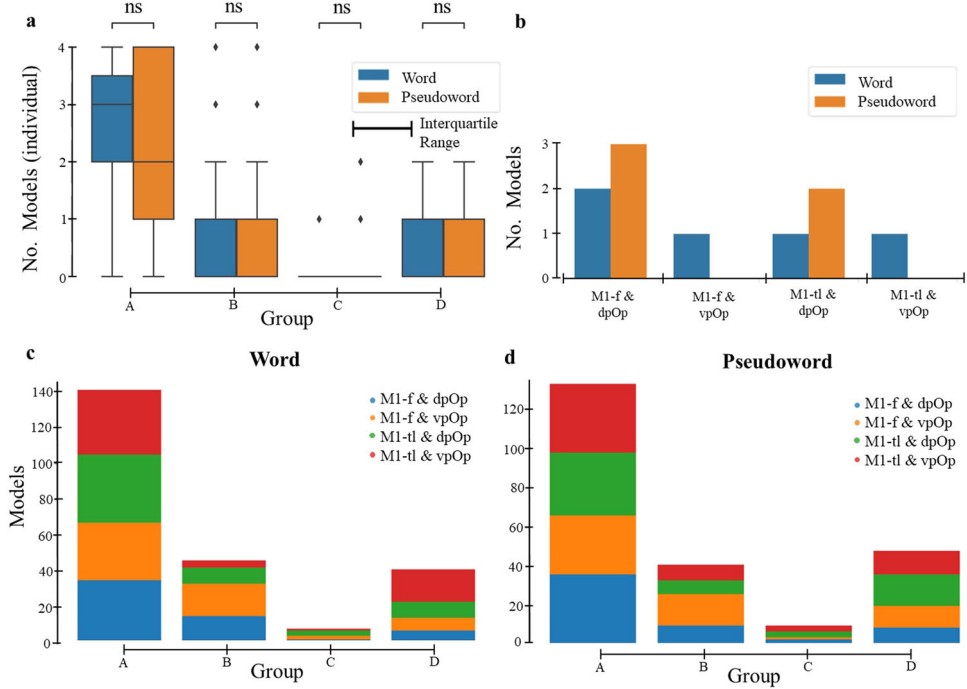

**Fig. 3 Model assignment in each group. a** Box plots of the number of models (y-axis) assigned to each group (x-axis), across the 4 configurations (dpOp & M1-f, vpOp & M1-f, dpOp & M1-tl, vpOp & M1-tl) for word and pseudoword repetition separately at an individual level. The box ranges from the first quartile to the third quartile of the distribution and the interquartile range represents the range between them. The line across the box is the median and the ends on the box plots go from the first and third quartiles to the most extreme data points. Here, Group A denotes excitatory connections from pOp to M1 and from pSTS to M1; Group B denotes excitatory connections from pSTS to M1 but not from pOp to M1; Group C denotes excitatory connections from pOp to M1 but not from pSTS to M1; and Group D denotes no connectivity from both pOp and pSTS to M1. We found no significant [ns] differences (with a p-value of 1.00) between word and pseudoword repetition across the different groups using a two-sided Mann-Whitney-Wilcoxon test with Bonferroni correction (sample size n = 59). **b** represents the 10 models (from subset of Group C) that were consistent with the neurological model across subregional configuration (dpOp & M1-f, vpOp & M1-f, dpOp & M1-tl, vpOp & M1-tl). **c, d** represent the model specification breakdown by group membership for word and pseudowords, respectively. This highlights the similar distribution of the 4 model types (in red, green, orange, and blue) for words and pseudowords and across subregional configuration.

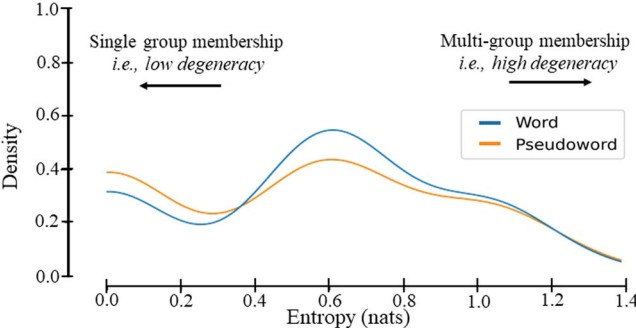

**Fig. 4 Degeneracy in auditory repetition.** Sample density (*y*-axis) is plotted against entropy measured in natural units (nats; *x*-axis), over group membership for word (blue) and pseudoword (orange) repetition separately. Low entropy (=0) denotes membership of a single group (e.g., the participant was consistently in group A) whereas high entropy (>1.3) denotes membership dispersed over all groups (i.e., the participant was in group A, B, C and D for the different subregional combinations). Sample density plots the participation distribution (*n* = 59) at each level of entropy.

participant variations in whether M1 was driven by pOp, pSTS or both and (iv) these results did not depend on whether the stimuli were words or pseudowords, pOp activity was extracted from ventral or dorsal aspects of pOp, or M1 activity was extracted from the face or tongue and larynx area. Below, we discuss the implications of each finding along with necessary directions for future experiments.

Positive (excitatory) connections from pSTS to M1 were observed during word repetition (in all but 3 participants) and in pseudoword repetition (in all but 8 participants). This effective connection is not consistent with the predictions of the neurological model and cannot be explained by lack of activity in pOp, which was consistently activated across participants—with most showing excitatory connectivity from pOp to M1: 39/59 during word repetition and 44/59 during pseudoword repetition. It was also surprising that connectivity from pSTS to pOp was inhibitory. Together, these observations provide evidence against the assumption of the neurological model that pOp mediates the influence of pSTS on M1.

Although not consistent with the neurological model, the separable influence of pSTS and pOp on M1 might explain (at least partially) why focal damage to pOp does not result in long-lasting speech production impairments[21–23]. Specifically, the connectivity from pSTS to M1 might be one of the key mechanisms underlying speech production recovery after pOp damage. Future fMRI studies of patients with relatively focal pOp damage[24] could therefore investigate the degree to which the effective connectivity from pSTS to M1 is related to preserved speech production abilities.

Our study does not elucidate which anatomical pathways sustain neuronal message passing from pSTS to M1. Theoretically, the supporting pathways may lie dorsal or ventral to the Sylvian fissure[25] and different pathways may be required for word and pseudoword repetition[26]. Previous studies have demonstrated a division of labour between the ventral and dorsal processing routes underlying language production[25–28]. A dorsal parietal-frontal stream is proposed to support form-to-articulation mapping (as needed for pseudoword repetition) and a ventral temporal–frontal stream is proposed to support form-to-meaning mapping[29] (as needed for word recognition). The general consensus is that word repetition, like pseudoword repetition, will be mediated, directly or indirectly by the dorsal pathway[9,20]. Nevertheless, future investigations are required to establish how different pathways, from pSTS, drive motor activity in M1.

Separately, the bidirectional inhibition between pOp and pSTS suggests that these regions sit at the same level of the cortical hierarchy for auditory speech repetition, with A1 below and M1 above. The argument for this functional heterarchy arises from the predictive processing assumption that forward connections (up the hierarchy) must be excitatory, and backward connections (down the hierarchy) are inhibitory or less excitatory[30–32]. More specifically, excitatory effective connectivity reflects prediction errors being passed from a lower to a higher area, whereas inhibitory effective connectivity reflects the explaining away (reduced excitation) of the prediction error lower in the hierarchy, generally thought to be mediated by inhibitory interneurons within the target region[33–36].

Given that pSTS is strongly associated with speech perception and pOp is strongly associated with the encoding of a speech plan (in both the 19th-century neurological model and 21st-century neuroscience), we propose that there may be turn-taking between pSTS and pOp. In other words, we propose reciprocal inhibition in terms of inferring what has been heard (excitation in pSTS) and what the participant is saying (excitation in pOp). This mutual inhibition is further endorsed by the phenomena of sensory attenuation; namely, the attenuation of self-produced sensations during speech[37–39].

We observed the following when considering the inter-participant and intra-participant variability in the connections to M1. First, the inter-participant variability analysis distinguished three hierarchical structures. The most common functional architecture (Group A) was a heterarchical organisation where pSTS and pOp can be thought of as superordinate to A1, with no clear hierarchical relationship between themselves. Conversely, Group B and Group C conformed to a hierarchical organisation. For Group B, this took the form of excitatory efferent connections from A1 to pSTS, and from pSTS to M1 and pSTS to pOp. Additionally, Group B featured inhibitory afferent connections from pOp to pSTS. From this, we infer that, under a predictive processing account, pSTS is lower in the functional hierarchy than pOp for Group B. Less than 5% of all models fell in Group C, where excitatory efferent connections are from A1 to pOp, pOp to M1, and pOp to pSTS: i.e., pSTS is hierarchically superordinate to pOp.

Only 2% of all models featured connectivity consistent with the classic pathway: i.e., excitatory connectivity from pSTS to pOp, and from pOp to M1 but not from pSTS to M1. However, even here, evidence for the classic pathway varied across task and subregional configurations. The inter- and intra-participant variability, across groups, illustrates degeneracy in functional architectures underwriting word repetition[17,19,40]. Participants might repeat words/pseudowords by either engaging pOp or using an alternative pathway involving pSTS. Moreover, the fact that individual participants moved from one hierarchical functional architecture to another, when repeating words or pseudowords, provides evidence that they were able to engage more than one processing route, whereas others preferred one over another.

This intra-participant variability has implications for how lesions to pSTS or pOp might change the effective connectivity of the network. Contrary to the neurological model, but consistent with current studies[27], we expect disconnections between pOp and pSTS—as a result of direct damage to pOp—to induce transitory auditory word repetition deficits. This is because damaging either region would mediate a readjustment of the overall effective connectivity. The resultant network should then be able to support auditory speech repetition. We plan to pursue this hypothesis in further work.

Briefly, in contradiction with classic and contemporary neurological models of auditory speech repetition, our results show that (i) pSTS drives M1 independently of pOp, (ii) there is

bilateral inhibitory connectivity between pOp and pSTS, and (iii) participants vary in the degree to which M1 activity is driven by pSTS or pOp. These findings (a) demonstrate a distributed, functional heterarchy between pSTS and pOp, (b) strongly imply alternative pathways for auditory speech repetition (degeneracy), and (c) serve to generate hypotheses about how auditory speech repetition can be maintained or recovered after brain damage. In addition, our findings strongly motivate further experiments to find the grey matter regions and white matter tracts that underlie the effective connectivity from pSTS to the primary motor cortex, bypassing pOp.

## Methods

**Participants**. A total of 59 participants were included in this study. Participant details are provided in Table 2. All participants were native English speakers, right-handed (assessed with the Edinburgh handedness inventory[41]) neurologically intact and reported normal or corrected-to-normal vision and hearing. The study was approved by the London Queen Square Research Ethics Committee and all relevant ethical regulations were followed. All participants gave written informed consent before participation and were compensated £10 per hour for their time.

**Experimental paradigm**. The current study focused on brain activation elicited when our participants were repeating heard words or pseudowords in different scanning runs (one run for word repetition and one run for pseudoword repetition). In each scanning run of 3.4 min, 40 words or pseudowords were presented sequentially with 4 blocks of 10 stimuli (25 s per block) interspersed with 16 s of rest (see Tables 3 and 4 for further details). The words were the following 40 object names: apple, banana, basket, biscuit, book, bread, cake, car, carrot, coconut, cup, duck, donkey, frog, grapes, guitar, kangaroo, king, koala, lizard, monkey, nun, pear, piano, pilot, plate, plane, potato, sack, shelf, snake, sofa, soldier, stool, swan, table, teapot, tortoise, truck, wineglass. They had an average of 1.68 syllables (range = 1–4) and an average duration of 0.65 s (standard deviation = 0.08 s). Pseudowords were the following 20 items presented twice but in

random order: enmich, fent, fint, hovet, irb, kig, kirs, lally, mox, numpy, nurry, seton, sutrid, thulo, touto, vaip, vum, wol, wox, zove. They had an average of 1.5 syllables (range = 1 to 4) and an average duration of 1.45 s (standard deviation = 0.15 s).

The auditory stimuli were presented using volume-adjusted MRI compatible headphones which filtered in-scanner noise. Before scanning, each participant was trained on how to correctly perform the task using a separate auditory stimulus set. During scanning, participants were instructed to respond immediately, whilst keeping as still as possible with their eyes open and fixated on a cross in the middle of the display screen. Their responses were recorded using a noise-cancelling MRI microphone and transcribed manually. An auditory repetition was marked as correct if it matched the target without any delays or self-correction.

In addition to word and pseudoword repetition, all participants performed 11 other conditions (one run per condition) that are not part of the current study (see Paradigm 2 in ref. [42]). Crucially, the order of all conditions, the content of the stimuli and the presentation parameters were identical for all participants, therefore inter-participant variability in brain activation cannot be explained by any of these factors.

**Data acquisition and analysis**. Functional MRI (fMRI) data were acquired on a 3 T Trio scanner (Siemens Medical Systems) using a 12-channel head coil and a gradient-echo EPI sequence with $3 \times 3$ mm in-plane resolution (repetition time/echo time/flip angle: 3080 ms/30 ms/90°, extended field of view = 192 mm, matrix size = $64 \times 64$, 44 slices, slice thickness = 2 mm, and interslice gap = 1 mm). Structural MRI data were high-resolution T1-weighted images, acquired on the same 3 T scanner using a 3D modified driven equilibrium Fourier transform sequence[43]: TR/TE/TI = 7.92 ms/2.48 ms/910 ms, Flip angle = 16, 176 slices, voxel size = $1 \times 1 \times 1$ mm3.

All data processing and analyses were performed with the Statistical Parametric Mapping (SPM12) software package (Wellcome Centre for Human Neuroimaging, London UK; http://www.fil.ion.ucl.ac.uk/spm/). All functional volumes were spatially realigned, unwarped, normalised to MNI space using a standard normalisation-segmentation procedure, and smoothed with a 6 mm full-width half-maximum isotropic Gaussian kernel, with a resulting voxel size of $3 \times 3 \times 3$ mm. The unwarping step corrects for distortions caused by head movement or magnetic field inhomogeneity. Within each scanning run, all participant's movements, were less than one voxel ($3 \times 3 \times 3$ mm).

The first level (fixed effects) analysis used the general linear model (GLM) to fit the pre-processed functional volumes for each of the 13 conditions (including word and pseudoword repetition). Separate regressors were entered for instructions, correct responses, incorrect responses and other responses (delayed, no response, or self-corrected). Each stimulus onset was modelled as a single event within each regressor. The contrasts of interest were those that modelled correct responses for word repetition compared to fixation and correct responses for pseudoword word repetition compared to fixation.

**Brain region selection**. The region of interest (ROI) selection process involved two steps. First, we defined the anatomical boundaries of each of our four regions of interest using the Brainnetome atlas[11] (Fig. 1a). We selected regions Te1.0 and Te1.2 for the primary auditory cortex (A1), the rostro-posterior STS subregion for Wernicke's area (pSTS), the dorsal pOp subregion for Broca's Area (dpOp), and the face (including the mouth) subregion for the primary motor cortex (M1-f). These choices were guided by the fMRI findings from an independent

| Table 3 Summary of experimental stimulus. | |
|---|---|
| **Stimuli** | |
| Word repetition | |
| Stimulus duration in sec (±std.) | 0.65 (0.08) |
| Average number of syllables (±std.) | 1.68 (0.73) |
| Pseudoword repetition | |
| Stimulus duration in sec (±std.) | 1.45 (0.15) |
| Average number of syllables (±std.) | 1.50 (0.51) |

| Table 4 Summary of experimental design specifications. | |
|---|---|
| **Experiment design** | |
| Number of sessions (50) for words | 1 |
| Number of sessions (50) for pseudowords | 1 |
| Number of blocks/run | 4 |
| Number of stimuli/block | 10 |
| Total number of stimuli per run | 40 |
| Inter stimulus interval (s) | 2.5 |
| Time per run (min) | 3.4 |
| TR (s) | 3.085 |
| Number of slices per volume | 44 |
| Number of volumes/run | 66 |
| Number of dummy acquisitions | 5 |

group of 25 neurologically intact participants who performed the same word and pseudoword repetition tasks as reported in ref. [5]. Additionally, we evaluated different parts of M1 and pOp due to (i) spatially extensive activation in these regions during auditory word and pseudoword repetition and (ii) lack of knowledge as to which parts of pOp were driving M1 and conversely which part of M1 was driven by pOp (or pSTS). For pOp, we exchanged the dorsal pOp subregion with the ventral pOp subregion (vpOp). For M1, we exchanged the face motor control subregion with the tongue and larynx motor control subregion (M1-tl) from the Brainnetome atlas. This resulted in 4 different (sub)regional configurations per subject for both word and pseudoword repetition (8 configurations per subject in total). We did not investigate different parts of A1 or pSTS because (i) we had strong a priori knowledge about the origin of auditory inputs in the primary auditory cortex and (ii) only the rostro-posterior part of STS was robustly activated by both word and pseudoword repetition.

The region borders were determined using a probability threshold of 50%: i.e., the anatomical localisation of the regions was consistent for at least 50% of the neurologically intact participants who contributed to the atlas construction. These probability thresholds are within the range used in previous studies[22,44–46].

Second, we searched for the peak response during word and pseudoword repetition within each anatomically defined ROI (Fig. 1a; Table 5) in each of the 59 participants. Separate time series of activation during the word and pseudoword repetition tasks were extracted from the peak coordinates for each participant. This ensured that effective connectivity between regions was estimated where activation was most robust for each participant, within a given ROI. In other words, we used each subject's functional anatomy to define ROI specific responses. In each region, group activation during both word and pseudoword repetition was significant at voxelwise $p < 0.05$ family-wise-error-corrected (using random field theory) for multiple comparisons across the whole brain (t-scores are reported in Table 5). Comparison of the coordinates for the peak response in each participant individually relative to that for the full group revealed that the mean distance was 0.74 mm (range = 0.37–1.56; standard deviation = 0.83).

**Dynamic causal modelling.** Effective connectivity among four ROIs was estimated using dynamic causal modelling (DCM)[15,47]

as implemented in SPM12. DCM is a hypothesis-driven framework for investigating models of effective connectivity in a network of interconnected neuronal populations using approximate Bayesian inference. It characterises the brain as a nonlinear dynamical system of interconnected neuronal populations whose directed connection strengths may be modulated by endogenous activity or external perturbations. Briefly, the model consists of a neuronal model, and a forward model, that describes how activity at the neuronal level translates into observed signals (Fig. 5a). DCM strives for a mechanistic explanation of experimental measures of brain activity in terms of directed intrinsic (self) and extrinsic (between region) connectivity. See ref. [12] for a detailed overview.

In the current study, we used two model parameters: (i) input parameters that identify which region was responding to external stimuli, here the primary auditory cortex; and (ii) the effective connectivity changes that occur among regions, as participants alternate between repetition and rest. These parameters were estimated at the neuronal level and the coupling between regions does not necessarily reflect the existence of direct (e.g., monosynaptic) connections.

For the technically savvy reader, we deliberately kept the analysis, and our interpretations, simple by separately estimating the average effective connectivity during each repetition task. In other words, our DCM models do not estimate the modulation of effective connectivity due to different experimental condition, i.e., word or pseudoword. Therefore, all interpretations of reported estimates should be read as average effective connectivity, and not the rate of change in the effective connectivity due to modulatory inputs.

**Participant-level DCM.** We now turn to the model specification. For each participant (and subregional configuration), we specified the model as defined in (Fig. 5b): (i) the driving input was from the primary auditory cortex (A1), (ii) A1 was connected to all regions except the primary motor cortex M1 (i.e., M1-f or M1-tl) given anatomical constraints, (iii) pSTS was connected to pOp (i.e., dpOp or vpOp), and (iv) M1 (i.e., M1-f or M1-tl) was defined as the output region that received inputs from either pSTS, pOp or both. Briefly, all specified connections were both forward and backward, and they could be either excitatory or inhibitory. Importantly, our specification formulated pSTS and pOp at a similar level in the structural hierarchy because they were both connected to the input in A1 and the output in M1. Moreover, this allowed us to estimate whether pOp was higher (or lower) than pSTS within the functional hierarchy.

We also specified an inhibitory self-connection for each region (which enables us to measure a region's sensitivity to its inputs). Changes in these self-connections can be regarded as a reflection of excitatory-inhibitory balance within each region[31]. The parameters are set to be negative (default is −0.5 Hz) to preclude run-away excitation in the network[12]. Accordingly, positive self-connection estimates are indicative of inhibition and negative self-connections are indicative of excitation.

All model parameters and their posterior probabilities were estimated, with Bayesian inversion, using variational Laplace[15], an automatic variational procedure under Gaussian assumptions about the form of the posterior. The participant-level specification was separately estimated for the different subregional configurations per participant for both word and pseudoword repetition, i.e., 8 DCM model estimations per participant in total.

**Group-level DCM.** We evaluated group effects and between-participant variability on parameters using the Parametric Empirical Bayes (PEB) model[13]. The resultant hierarchical model

**Table 5 Effects reported have been thresholded at voxel wise _p_ < 0.05 FWE-corrected.**

| Brain region | Peak co-ordinate | | | T-score | Extent |
|---|---|---|---|---|---|
| | x | y | z | | |
| _Word repetition_ | | | | | |
| M1-f | −48 | −13 | 38 | 23.21 | 198v |
| M1-tl | −57 | 2 | 2 | 12.78 | 87v |
| pSTS | −57 | −31 | 5 | 17.15 | 85v |
| dpOp | −42 | 8 | 26 | 10.25 | 100v |
| vpOp | −51 | 8 | −1 | 10.47 | 48v |
| A1 | −51 | −19 | 8 | 20.85 | 234v |
| _Pseudoword repetition_ | | | | | |
| M1-f | −48 | −13 | 38 | 21.77 | 193v |
| M1-tl | −54 | 5 | −1 | 11.04 | 63v |
| pSTS | −57 | −31 | 5 | 18.31 | 75v |
| dpOp | −39 | 5 | 26 | 12.68 | 101v |
| vpOp | −51 | 8 | −1 | 8.77 | 42v |
| A1 | −51 | −19 | 8 | 19.97 | 220v |

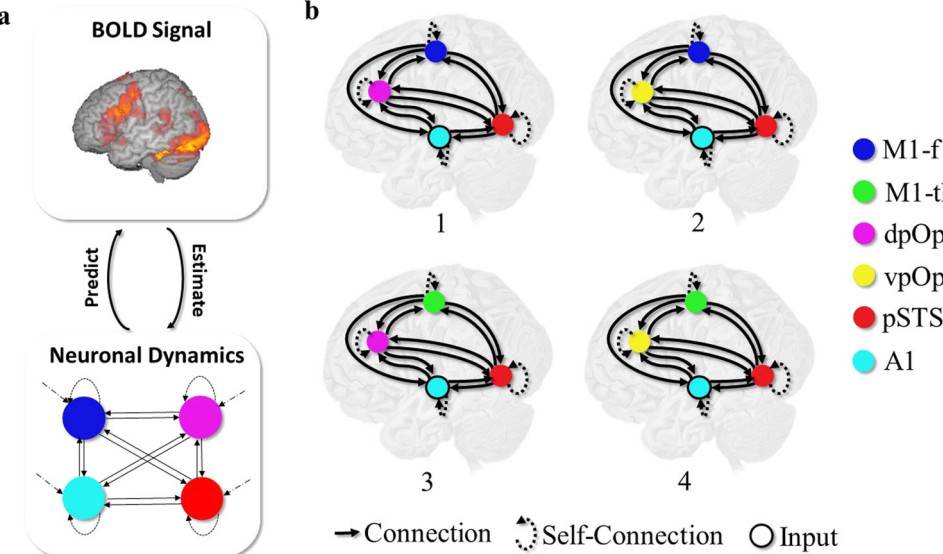

**Fig. 5 Graphical illustration of subject-level DCM. a** presents a graphic illustration of the DCM for generative model: BOLD (blood oxygenation level-dependent) signal represents the observed fMRI data, Neuronal Dynamics represent the neural state dynamics and arrows between denote the forward model (i.e., predict) and inverse model (i.e., estimate). **b** presents the four different DCMs estimated. Each model comprised 4 regions with 15 connections, including the 4 self-connections. However, different subregional configurations lead to 4 different models (1-4): 1 (M1-f and dpOp), 2 (M1-f and vpOp), 3 (M1-tl and dpOp) and 4 (M1-tl and dpOp). Here, dark blue coloured node is for M1-f, green for M1-tl, magenta for dpOp, yellow for vpOp, red for pSTS and cyan for A1.

quantifies the estimated connection strengths, and their uncertainty, from the participant to the group level. Having estimated the group-level parameters (e.g., group-average effective connection strengths), we used Bayesian model comparison to test hypotheses for alternative models of effective connectivity during word and pseudoword repetition separately. The alternative models were generated by switching parameters on and off using an automatic grid search[13]. Explicitly, the group-level DCM estimates were evaluated across the 8 subregional configurations (4 for word repetition and 4 for pseudoword repetition). For each configuration, we evaluated the model evidence across 256 separate models where each model represented the removal of a particular parameter or effective connection e.g., from A1 to pSTS or from pSTS to M1. Finally, the model with the highest model evidence was selected for each of the 8 subregional configurations.

**Statistics and reproducibility**. All analyses were performed using SPM software in MATLAB and no modifications were made to the underlying methods. Technical details with specific parameters are described in this section. Briefly, the first level (fixed effects) analysis used a GLM to fit the pre-processed functional volumes for each of the 13 conditions controlled for instructions, correct responses, incorrect responses, and other responses for the 59 subjects. We used the following contrasts for voxel-based analysis to localise ROI per subject given pre-defined anatomical regions: correct responses for word repetition compared to fixation and correct responses for pseudoword word repetition compared to fixation. Euclidean mean distance was used to evaluate the peak response (ROI localisation) consistency across individuals (within 0.74 mm with a range = 0.37–1.56; standard deviation = 0.83). Extracted ROI timeseries were modelled using DCM (as specified in SPM12[48]) i.e., variational Laplace approximation used for fitting the data to the fMRI DCM state-space model. Statistical differences between individual-level estimated effects were evaluated using a two-sided Mann-Whitney-Wilcoxon test with Bonferroni correction (sample size $n = 59$).

Estimates from the first-level DCM were used to fit the second-level PEB model.

**Reporting summary**. Further information on research design is available in the Nature Portfolio Reporting Summary linked to this article.

## Data availability
The source data behind the graphs in Figs. 3 and 4 are provided in Supplementary Data 2. The source data behind the graphs in Fig. 3 are provided in Supplementary Data 3. Any other data are available via request to c.j.price@ucl.ac.uk.

## Code availability
We used generic functions to model our DCM. This is available in MATLAB as part of the SPM academic software[48]: http://www.fil.ion.ucl.ac.uk/spm/.

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

## Acknowledgements

The authors would like to thank Shamima Khan and PLORAS team for their efforts in supporting the data collection. Data collection was funded by the Wellcome Centre for Human Neuroimaging (Ref: 203147/Z/16/Z). N.S. is funded by the Medical Research Council (MR/S502522/1) and the 2021-2022 Microsoft PhD Fellowship. T.M.H. is funded by the Stroke Association (TSA_PDF_2017/02). C.J.P. is funded by the Wellcome Trust (205103/Z/16/Z), the Medical Research Council (MR/M023672/1) and Stroke Association (TSA 2014/02). A.G-V. and D.L.L.-P. were each supported by a postdoctoral fellowship from the Chilean National Agency for Research and Development (ANID BECAS-CHILE 74200065 and ANID BECAS-CHILE 74200073, respectively). D.L.L.-P. is supported with funding from the Chilean National Agency for Research and Development (ANID SUBVENCIÓN A LA INSTALACIÓN EN LA ACADEMIA 85220006).

## Author contributions

Conceptualisation: N.S., A.G.V., K.F., C.P. Experimental Design: C.P. Data Collection and Processing: A.G.V., D.L.P., J.E., T.H., C.P. Formal analysis: N.S. Methodology: N.S., A.G.V., K.F., C.P. Supervision: K.F., C.P. Visualisation: N.S., A.G.V., J.E. Writing—original draft: N.S., C.P. Writing—review & editing: N.S., A.G.V., J.E., D.L.P., D.G., T.H., K.F., C.P.

## Competing interests

The authors declare no competing interests.

## Additional information



