## [Peer review file · Communications Biology]

Degeneracy in the neurological model of auditory speech repetitionReviewers' comments:

Reviewer #1 (Remarks to the Author):

The manuscript of Sajid et al. reports human neuroimaging data while participants performed a speech repetition task. The authors tested a standard neurological model for speech repetition against an empirical analysis of effective brain connectivity. Brain connectivity was quantified as functional connectivity between 4 brain nodes in the left hemisphere that are central to speech repetition. The empirical data surprisingly contract many assumptions of brain connectivity as proposed by the standard neurological model, including unexpected inhibitory connection and connectivity between brain nodes that were not taken into account before.

Overall, I like the manuscript, the study, and the type of data analysis very much. I especially like this empirical approach that violates long-lasting assumptions from some rather outdated speech processing models. The paper seems thus a candidate for publication in Communication Biology, given that the author addresses several major points.

(1) In the abstract (last sentence) and throughout the manuscript the authors talk about the "degeneracy" of the standard model. Although I think I know what the authors mean here, it's not entirely clear to me- Please introduce the concept of degeneracy more clearly.

(2) The Introduction might focus too much on only the standard model. I would like to read also about alternative views and data, that have been reported by previous studies. Are the findings by the authors completely unexpected, or are there previous studies, which already point to some alternatives to the standard model? A more balanced summary of the previous literature might be good here.

(3) The results section directly jumps into reporting the data. It might be good to give a short summary of the analysis approach first, because the methods are reported at the end of the manuscript.

(4) A major shortcoming of the manuscript is the missing description of the experimental task and details. How were the words/pseudowords presented? And how were the participants instructed to repeat the words? How did the authors make sure that repetitions were correct? What words/pseudowords were used? How were the trials modeled in the first-level statistical model? Did the authors control for head movements? Etc. Without this information, it is hard to estimate of the DCM findings are valid and accurately modeled. The manuscript thus needs to be re-assessed with this missing information included.

(5) Fig. 1: the arrows are hard to see.

Reviewer #2 (Remarks to the Author):

The study sets to examine the predictions of the 19th century neurological model that features serial connectivity from "Wernicke" to "Broca's" areas, and from "Broca" to M1. Using an auditory repetition task in fMRI and an effective connectivity analysis tool (DCM) they show that the anatomical regions chosen to represent these historical labels are in fact not connected serially but rather reciprocally for most individuals, and that the connectivity patterns varies across individuals, supporting the notion of degeneracy. They also show that the connectivity pattern within individuals varies depending on the choice of different anatomical sub-regions within "Broca" or M1.

This is an interesting study with a sound methodology. I have few concerns that are mostly related to the study rationale and the justification of certain elements of the methods and results.

Introduction:

1. As the authors acknowledge in the first paragraph, there has long been mounting evidence against the simplicity of the 19th century model, in terms of the sheer number of brain regions involved in the tasks of speech perception and speech production, beyond those included in this model. Although the authors cite Hope et al. (2014) for why they still focus on this minimal set of regions, that paper does not seem to narrow the set of critical regions to the task of auditory word repetition. Similarly, the hypothesis for a serial connectivity is also one that can only be justified based on a historical perspective, as very few brain networks studied by modern tools are found to

work serially. This is to say that it should be made clear from the beginning that the reasons to test the predictions of this model are merely historical, rather than as an actively accepted model for how auditory repetitions are carried out in the brain.

2. The rationale for using the 4 configurations (two anatomical sub-regions within two of the larger areas), is not clear. A. The authors say this was done "To ensure that inter-subject variability was not a consequence of the precise anatomical definition of regions.."(line 66). However, to test this one would need to choose a winning model out of the 4 configurations for each individual, and look specifically at its connectivity pattern. B. The authors do not provide justification for enabling anatomical variability (two optional location) for pOp and M1 and not for pSTS.

3. When the authors talk about "intra-subject variability" (e.g. lines 64-66): this may imply changes across time points, whereas in practice they refer to changes across the different configurations of sub-regions. A. This should be clarified. B. It is not clear what is the justification for the hypothesis that different anatomical sub-regions should show the same pattern of connectivity.

Results

4. The wording on Line 93: "The differences in effectivity connectivity for pseudoword repetition, compared to word repetition" should be modified to clarify that this is just a conceptual comparison, rather than a direct comparison, (as words and pseudowords were not modelled together and could not be compared).

5. Figure 2: There is no legend for the color of the dots. Are these different configurations of subregions in pOp and M1? The distribution of the red and green dots seems to be different from that of the blue and orange dots particularly for group A, for both words and pseudowords. Is there a way to assess this difference and interpret it?

6. Figure 2: The figure seems to show all individual models, without any thresholding (e.g. only connections with posterior probability >0.75). However, the groups (A-D) are defined based on the existence/ absence of excitatory connections. This seems to imply that individual models where there are excitatory/ inhibitory connections with low posterior probability should not be shown in the figure or in the subsequent analysis. What is the justification for not applying any threshold?

7. Figure 4 – It is unclear how to interpret this figure, as most individuals are in the middle, and it is unclear what level of entropy is supportive of degeneracy and why. The same goes for the description (Line 175), where the numbers (0.49, and 0.55) are presented without an explanation of the scale.

Typos and minor comments:

8. Figure 3: y-axis should be "mean number of models per individual". In the legend: "average across 8 configurations" is probably wrong (should be 4), because words and pseudowords are depicted separately.

9. "Effectivity connectivity" should be "effective connectivity" in various places.

10. The information given in Table 2 does not benefit from a table format.

11. Methods: The number of items of each type (words and pseudowords) are only presented per run, but it is not clarified how many runs for each stimuli type were included.

12. Fig 5a – the lower box is confusing as it does not represent the models used in the current study.

13. Line 282 – 'underwriting' should be 'underlying'.

Re: COMMSBIO-22-1537-T

Dear Dr. George Inglis and Reviewers,

We would like to thank you for reviewing our manuscript and for providing positive and constructive feedback. The reviewers have raised some excellent points, and we have used these comments to improve the manuscript and highlight its significance.

We address each comment in detail below: We have reproduced reviewers' comments and editorial comments and appended our responses directly below. We have included revisions (**in bold**) to the manuscript for your convenience.

We hope that these changes are what you and the reviewers had in mind.

With very best wishes,

Noor Sajid, Andrea Gajardo-Vidal, Justyna O Ekert, Diego L Lorca-Puls, Thomas MH Hope, David W Green, Karl J Friston, and Cathy J Price

Reviewer #1

The manuscript of Sajid et al. reports human neuroimaging data while participants performed a speech repetition task. The authors tested a standard neurological model for speech repetition against an empirical analysis of effective brain connectivity. Brain connectivity was quantified as functional connectivity between 4 brain nodes in the left hemisphere that are central to speech repetition. The empirical data surprisingly contract many assumptions of brain connectivity as proposed by the standard neurological model, including unexpected inhibitory connection and connectivity between brain nodes that were not taken into account before.

Overall, I like the manuscript, the study, and the type of data analysis very much. I especially like this empirical approach that violates long-lasting assumptions from some rather outdated speech processing models. The paper seems thus a candidate for publication in Communication Biology, given that the author addresses several major points.

Thank you.

(1) In the abstract (last sentence) and throughout the manuscript the authors talk about the "degeneracy" of the standard model. Although I think I know what the authors mean here, it's not entirely clear to me- Please introduce the concept of degeneracy more clearly.

To address this, we have defined "degeneracy" as follows:

Abstract (Line 23-26): "These results suggest that there are different neural pathways that can support auditory speech repetition. This "degeneracy" may explain resilience to functional loss after brain damage. Degeneracy is used here in the sense of degenerate (i.e., many-to-one) structure-function relationships. In other words, several neural pathways or architectures can fulfil the same function.

Introduction (Line 80-83): *“We were also interested in.... whether there was evidence for degeneracy in functional architectures; namely, whether effective connectivity varied across participants (inter-subject variability) or within participants (intra-subject variability) for the same task.”*

Introduction (Line 83-87): *“Evidence for degeneracy in terms of variation within and between subjects, would provide insights into how language functions recover following neurological damage. Under this formulation, structures (e.g., subgraphs of a neuronal network) may be sufficient, but not necessary, for a particular function—meaning that functional deficits arise only when all degenerate neural pathways are damaged^{15, 16, 17}.”*

Results (Line 203-207): *“Degeneracy pertains to how many independent structures or neural pathways can be recruited to subserve the same function or task^{15, 16, 20}, for example, auditory repetition of heard speech. In our results, degeneracy in the neural pathways supporting auditory speech repetition is implied by the high inter- and intra-subject variability in group membership, i.e., variability in the degree to which M1 activity was driven by pSTS, pOp, both or neither.”*

Results (Line 226-228): *“... and the majority (73%) of our participants were assigned to at least 2 different groups, indicating that they could execute the repetition tasks in different ways (i.e., “degeneracy”).”*

Discussion (Line 315-317): *“The inter- and Intra-participant variability, across groups, illustrates degeneracy in functional architectures underwriting word repetition^{17, 20, 41}”*

Conclusion (Line 332-335): *“These findings (a) demonstrate a distributed, functional heterarchy between pSTS and pOp, (b) strongly imply alternative pathways for auditory speech repetition (degeneracy), and (c) serve to generate new hypotheses about how auditory speech repetition can be maintained or recovered after brain damage.”*

(2) The Introduction might focus too much on only the standard model. I would like to read also about alternative views and data, that have been reported by previous studies. Are the findings by the authors completely unexpected, or are there previous studies, which already point to some alternatives to the standard model? A more balanced summary of the previous literature might be good here.

Thank you for highlighting this. Our results are indeed completely unexpected, according to both classic and contemporary models of word repetition. We have clarified this, throughout the manuscript:

Abstract (Line 12): *“Both classic and contemporary models of auditory word repetition involve at least four left hemisphere regions:”*

Introduction (Line 42-46): *“More contemporary models of auditory speech repetition recognize the existence of direct and indirect connections between parts of Wernicke’s area and Broca’s area, with the indirect pathway routed via the temporo-parietal junction (also referred to as the sylvian parietal temporal area, Spt) and/or the supramarginal gyrus^{9, 10}.”*

Introduction (Line 76-78): *“If the classic and contemporary models of auditory word repetition are correct, our DCM analysis should show that activity in the primary auditory cortex excites pSTS, pSTS excites pOp and pOp excites the primary motor cortex.”*

Discussion (Line 245-251): *“Contemporary models of auditory speech repetition incorporate greater anatomical precision (e.g., here we use pSTS as a proxy for Wernicke’s area and pOp as a proxy for Broca’s area) and posit direct and indirect connections between these temporal and frontal areas, due to the involvement of other regions (e.g., in temporo-parietal cortex and subcortical structures). However, the general consensus across classic and contemporary models is that pSTS connects directly or indirectly with M1 through pOp (and other regions)^{9, 21}.”*

Conclusions (Line 331-334): *“In contradiction with classic and contemporary neurological models of auditory speech repetition, our results show that (i) pSTS drives M1 independently of pOp, (ii) there is bilateral inhibitory connectivity between pOp and pSTS, and (iii) participants vary in the degree to which M1 activity is driven by pSTS or pOp. “*

(3) The results section directly jumps into reporting the data. It might be good to give a short summary of the analysis approach first, because the methods are reported at the end of the manuscript.

Thank you. We made two changes to improve the clarity of the results. Summary of methods added to Introduction:

(Line 52-75) “The current paper investigates how Wernicke’s and Broca’s areas interact with one another during auditory speech repetition, and how they are driven by, and/or drive responses in left auditory cortex (A1), and the left primary motor cortex (M1). Our prior fMRI studies of auditory word and pseudoword repetition⁵ identified activation in: dorsal and ventral parts of the pars opercularis (dpOp and vpOp) in the vicinity of Broca’s area, the rostro-posterior superior temporal sulcus (pSTS), in the vicinity of Wernicke’s area, primary auditory cortex (A1) and motor cortex (M1) including both the face region (M1-f) and the tongue and larynx region (M1-tl) in addition to other areas that are not under investigation here. To ensure accurate anatomical localisation, regions of interest were defined using the Brainnetome atlas¹¹ (Figure 1a), and the time series of activation during auditory word and pseudoword repetition was extracted from the peak voxel within each anatomically defined region of interest for 59 neurotypical participants. Considering the two subdivisions of pOp (dpOp and vpOp)—and of M1 (M1-f and M1-tl)—there were four possible regional configurations that include A1 and pSTS with either: (1) dpOp and M1-f, (2) vpOp and M1-f, (3) dpOp and M1-tl, and (4) vpOp and M1-tl.

Directed interactions (i.e., effective connectivity) across activity in A1, pSTS, pOp and M1, during auditory repetition, was assessed using Dynamic Causal Modelling (DCM)^{11, 12, 13, 14}. Contrary to the classic neurological model, we did not assume a serial relationship among the four regions but tested for bilateral connections between all regions, except the input (A1) and output (M1) regions. Furthermore, our analyses were agnostic as to the white matter tracts mediating effective connectivity. For example, effective connectivity from pSTS to pOp would not necessarily imply monosynaptic connections via a direct white matter tract (e.g., the arcuate fasciculus). It could also be the consequence of indirect (polysynaptic) connections (e.g., via Spt or SMG) or even multiple fasciculi. In other words, effective connectivity can be mediated vicariously through intervening cortical stations (not included in the model).”

Replacing the first paragraph of the results with:

(Line 91-99) "Effective connectivity was assessed with Bayesian model comparison – which compares the evidence for models with and without each connection. The strength of connection is quantified in terms of posterior estimates of the model's connectivity parameters. For extrinsic (i.e., between region) connections, a positive connection is excitatory (i.e., activity in one region increases activity in another), while a negative connection is inhibitory (i.e., activity in one region decreases activity in another). For intrinsic (within region) connections, positivity means that increased activity results in greater self-inhibition, whereas negativity means increased activity results in less self-inhibition. Throughout the results, tables, and figures, we only report estimated connectivity with a posterior probability greater than 0.75."

(4) A major shortcoming of the manuscript is the missing description of the experimental task and details. How were the words/pseudowords presented? And how were the participants instructed to repeat the words? How did the authors make sure that repetitions were correct? What words/pseudowords were used? How were the trials modeled in the first-level statistical model? Did the authors control for head movements? Etc. Without this information, it is hard to estimate if the DCM findings are valid and accurately modeled. The manuscript thus needs to be re-assessed with this missing information included.

We apologise for this oversight and assume that text was deleted when formatting the paper for submission. The missing information has now been included in the methods section, as follows:

How were the words/pseudowords presented?

(Line 365-367) "The auditory stimuli were presented using volume-adjusted MRI compatible headphones which filtered in-scanner noise. Before scanning, each participant was trained on how to correctly perform the task using a separate auditory stimulus set."

How were the participants instructed to repeat the words?

(Line 367-369) "During scanning, participants were instructed to respond immediately, whilst keeping as still as possible with their eyes open and fixated on a cross in the middle of the display screen."

How did the authors make sure that repetitions were correct?

(Line 369-371) "Their responses were recorded using a noise-cancelling MRI microphone and transcribed manually. An auditory repetition was marked as a correct if it matched the target without any delays or self-correction."

What words/pseudowords were used?

(Line 356-364) "The words were the following 40 object names: apple, banana, basket, biscuit, book, bread, cake, car, carrot, coconut, cup, duck, donkey, frog, grapes, guitar, kangaroo, king, koala, lizard, monkey, nun, pear, piano, pilot, plate, plane, potato, sack, shelf, snake, sofa, soldier, stool, swan, table, teapot, tortoise, truck, wineglass. They had an average of 1.68 syllables (range = 1 to 4) and an average duration of 0.65s (standard deviation = 0.08s). Pseudowords were the following 20 items presented twice but in random order: enmich, fent, fint, hovet, irb, kig, kirs, lally, mox, numpy, nurry, seton, sutrid, thulo, touto, vaip, vum, wol, wox, zove. They had an average of 1.5 syllables (range = 1 to 4) and an average duration of 1.45s (standard deviation = 0.15s)."

How were the trials modelled in the first-level statistical model?

(Line 393-399) "The first level (fixed effects) analysis used the general linear model (GLM) to fit the pre-processed functional volumes for each of the 13 conditions (including word and pseudoword repetition). Separate regressors were entered for instructions, correct responses, incorrect responses and "other" responses (delayed, no response, or self-corrected). Each stimulus onset was modelled as a single event within each regressor. The contrasts of interest were those that modelled correct responses for word repetition compared to fixation and correct responses for pseudoword word repetition compared to fixation."

Did the authors control for head movements?

(Line XX-XX) "All functional volumes were spatially realigned, unwarped, normalised to MNI space using a standard normalisation-segmentation procedure, and smoothed with a 6 mm full-width half-maximum isotropic Gaussian kernel, with a resulting voxel size of 3 × 3 × 3 mm. The unwarping step corrects for distortions caused by head movement or magnetic field inhomogeneity. Within each scanning run, all participant's movements, were less than one voxel (3 × 3 × 3 mm)."

(5) Fig. 1: the arrows are hard to see.

Thank you, we have now made the arrows bigger. Please refer to Figure 1 in the manuscript.

Reviewer #2

The study sets to examine the predictions of the 19th century neurological model that features serial connectivity from "Wernicke" to "Broca's" areas, and from "Broca" to M1. Using an auditory repetition task in fMRI and an effective connectivity analysis tool (DCM) they show that the anatomical regions chosen to represent these historical labels are in fact not connected serially but rather reciprocally for most individuals, and that the connectivity patterns varies across individuals, supporting the notion of degeneracy. They also show that the connectivity pattern within individuals varies depending on the choice of different anatomical sub-regions within "Broca" or M1.

Thank you for the excellent precis.

This is an interesting study with a sound methodology. I have few concerns that are mostly related to the study rationale and the justification of certain elements of the methods and results.

We hope our revisions are what you had in mind. Please let us know if you have any further concerns.

Introduction:

1. As the authors acknowledge in the first paragraph, there has long been mounting evidence against the simplicity of the 19th century model, in terms of the sheer number of brain regions involved in the tasks of speech perception and speech production, beyond those included in this model. Although the

authors cite Hope et al. (2014) for why they still focus on this minimal set of regions, that paper does not seem to narrow the set of critical regions to the task of auditory word repetition.

Similarly, the hypothesis for a serial connectivity is also one that can only be justified based on a historical perspective, as very few brain networks studied by modern tools are found to work serially.

Thank you. We agree and acknowledge this with the following:

Abstract (Line 15-18): *“Previous functional-MRI (fMRI) studies confirm that auditory repetition activates these regions, in addition to many others. Crucially, however, contemporary models do not specify how regions interact and drive each other during auditory repetition.”*

Introduction (Line 67-69): *“Contrary to the classic neurological model, we did not assume a serial relationship among the four regions but tested for bilateral connections between all regions except the input (A1) and output (M1) regions.”*

2. The rationale for using the 4 configurations (two anatomical sub-regions within two of the larger areas), is not clear. A. The authors say this was done “To ensure that inter-subject variability was not a consequence of the precise anatomical definition of regions.”(line 66). However, to test this one would need to choose a winning model out of the 4 configurations for each individual and look specifically at its connectivity pattern. B. The authors do not provide justification for enabling anatomical variability (two optional location) for pOp and M1 and not for pSTS.

Thank you for noting an error here. We have added the following text to clarify.

Methods (Line 407-417): *“Additionally, we evaluated different parts of M1 and pOp due to (i) spatially extensive activation in these regions during auditory word and pseudoword repetition and (ii) lack of knowledge as to which parts of pOp were driving M1 and conversely which part of M1 was driven by pOp (or pSTS). For pOp, we exchanged the dorsal pOp subregion with the ventral pOp subregion (vpOp). For M1, we exchanged the face motor control subregion with the tongue and larynx motor control subregion (M1-tl) from the Brainnetome atlas. This resulted in 4 different (sub)regional configurations per subject for both word and pseudoword repetition (8 configurations per subject in total). We did not investigate different parts of A1 or pSTS because (i) we had strong a priori knowledge about the origin of auditory inputs in the primary auditory cortex and (ii) only the rostro-posterior part of STS was robustly activated by both word and pseudoword repetition.”*

Introduction (Line 54-65): *“Our prior fMRI studies of auditory word and pseudoword repetition⁵ identified activation in: dorsal and ventral parts of the pars opercularis (dpOp and vpOp) in the vicinity of Broca’s area, the rostro-posterior superior temporal sulcus (pSTS), in the vicinity of Wernicke’s area, primary auditory cortex (A1) and motor cortex (M1) including both the face region (M1-f) and the tongue and larynx region (M1-tl) in addition to other areas that are not under investigation here. To ensure accurate anatomical localisation, regions of interest were defined using the Brainnetome atlas¹¹ (Figure 1a), and the time series of activation during auditory word and pseudoword repetition was extracted from the peak voxel within each anatomically defined region of interest for 59 neurotypical participants. Considering the two subdivisions of pOp (dpOp and vpOp)—and M1 (M1-f and M1-tl)—there were four possible regional configurations that include A1 and pSTS with either: (1) dpOp and M1-f, (2) vpOp and M1-f, (3) dpOp and M1-tl, and (4) vpOp and M1-tl.*

3. When the authors talk about “intra-subject variability” (e.g., lines 64-66): this may imply changes across time points, whereas in practice they refer to changes across the different configurations of sub-regions. A. This should be clarified. B. It is not clear what is the justification for the hypothesis that different anatomical sub-regions should show the same pattern of connectivity.

Thank you for highlighting this. Regarding (A), we have clarified as follows:

(Line 203-210): “Degeneracy pertains to how many independent structures or neural pathways can be recruited to subserve the same function or task^{15, 16, 20}, for example, auditory repetition of heard speech. In our results, degeneracy in the neural pathways supporting auditory speech repetition is implied by the high inter- and intra-subject variability in group membership, i.e., variability in the degree to which M1 activity was driven by pSTS, pOp, both or neither. Inter-participant variability in connectivity, within the same task and within the same regional configuration, indicates that there are different ways that the same task can be performed across subjects.”

(Line 215-220) Intra-participant variability in connectivity was only assessed across tasks and regional configurations. It illustrates that participants have the capacity to use different neural pathways. Theoretically, such variation could be the consequence of task or regional configuration. However, this is unlikely because (i) inter-patient variability was observed when task and regional configuration was controlled and (ii) there were no consistent differences in which connections were used for words and pseudowords (see Figure 3).”

Regarding (B). We do not assume that different anatomical subregions should show the same pattern of connectivity. This was an observation from the data at the group level. At the individual level, there is variation in how the regions were recruited but there was no consistent pattern in how connections were recruited across participants. This should now be clearer from the revised Figure 3.

Results

4. The wording on Line 93: “The differences in effectivity connectivity for pseudoword repetition, compared to word repetition” should be modified to clarify that this is just a conceptual comparison, rather than a direct comparison, (as words and pseudowords were not modelled together and could not be compared).

Good point. We have reworded as:

(Line 114-115): “In contrast to word repetition, effective connectivity for pseudoword repetition identified”.

5. Figure 2: There is no legend for the color of the dots. Are these different configurations of subregions in pOp and M1? The distribution of the red and green dots seems to be different from that of the blue and orange dots particularly for group A, for both words and pseudowords. Is there a way to assess this difference and interpret it?

Thank you for highlighting this. We have updated the description in the figure legend:

(Line 181-182) The colour dots denote the model specification, i.e., blue is for the M1-f and dpOp, orange for M1-f and vpOp, green for M1-tl and dpOp and red for M1-tl and vpOp.

Furthermore, we have extended Figure 3 to show the distribution of each regional configuration in each connectivity group.

6. Figure 2: The figure seems to show all individual models, without any thresholding (e.g., only connections with posterior probability >0.75). However, the groups (A-D) are defined based on the existence/ absence of excitatory connections. This seems to imply that individual models where there are excitatory/ inhibitory connections with low posterior probability should not be shown in the figure or in the subsequent analysis. What is the justification for not applying any threshold?

Apologies for the confusion. All results were thresholded with the same posterior probability. We clarify this in the last sentence of the first paragraph of the results as follows:

(Line 91-99) "Effective connectivity was assessed with Bayesian model comparison – which compares the evidence for models with and without each connection. The strength of connection is quantified in terms of posterior estimates of the model's connectivity parameters. For extrinsic (i.e., between region) connections, a positive connection is excitatory (i.e., activity in one region increases activity in another), while a negative connection is inhibitory (i.e., activity in one region decreases activity in another). For intrinsic (within region) connections, positivity means that increased activity results in greater self-inhibition, whereas negativity means increased activity results in less self-inhibition. Throughout the results, tables, and figures, we only report estimated connectivity with a posterior probability greater than 0.75."

We also clarified the assignment of models to different groups as follows:

(Line 152-154) "Each model was assigned to one of four groups (A-D in Figure 2), according to the presence or absence of significant of excitatory connectivity from pOp or pSTS to M1 (posterior probability >0.75)."

(Line 157-158) "Group D was defined by the absence of definitive connections from both pOp and pSTS to M1 (i.e., a posterior probability range of <0.75)."

7. Figure 4 – It is unclear how to interpret this figure, as most individuals are in the middle, and it is unclear what level of entropy is supportive of degeneracy and why. The same goes for the description (Line 175), where the numbers (0.49, and 0.55) are presented without an explanation of the scale.

Thank you for highlighting the lack of clarity here. We have added the following text to provide the details:

(Line 206-231) "In our results, degeneracy in the neural pathways supporting auditory repetition is implied by the high inter- and intra-subject variability in group membership, i.e., variability in the degree to which M1 activity was driven by pSTS, pOp, both or neither. Inter-participant variability in connectivity, within the same task and within the same regional configuration, indicates that there are different ways that the same task can be performed across subjects. Intra-participant variability in connectivity was only assessed across tasks and regional configurations. In other words, we assessed intra-subject

variability as the number of connectivity groups (1-4) that each participant was assigned to. For example, during word repetition, subject C073 belonged to both Group B (3x) and Group C (1x). Details for all participants are shown in Supplementary Table 1.”

“Intra-participant variability illustrates that participants have the capacity to use different neural pathways. Theoretically, such variation in the current study could be the consequence of task or regional configuration. However, this is unlikely because (i) inter-patient variability was observed when task and regional configuration was controlled and (ii) there were no consistent differences in which connections were used for words and pseudowords (see Figure 3). In information theory, the amount of uncertainty or randomness in a system can be quantified as entropy and measured in units of nats, which is a logarithmic scale based on the natural logarithm with a base of e . This is particularly useful when dealing with probabilities that are not evenly distributed and has previously been used to denote degeneracy²⁰ In our results, low entropy (=0) denotes membership of a single group (e.g., the participant was consistently in group A) whereas high entropy (>1.3) denotes membership dispersed over all groups (i.e., the participant was in group A, B, C and D for the different regional combinations or tasks). We found that the average entropy in individual group membership was 0.49 for word repetition and 0.55 for pseudoword repetition (Figure 4) and the majority (73%) of our participants were assigned to at least 2 different groups, indicating that they could execute the repetition tasks in different ways (i.e., “degeneracy”).”

Typos and minor comments:

8. Figure 3: y-axis should be “mean number of models per individual”. In the legend: “average across 8 configurations” is probably wrong (should be 4), because words and pseudowords are depicted separately.

Thank you - good spot. We have now updated this Figure to show the number of models (y axis) assigned to each group (x axis), for each of the 4 configurations (dvpOp/M1-f, vpOp/M1-f, dpOp/M1-tl, vpOp/M1-tl) for word and pseudoword repetition separately.

9. “Effectivity connectivity” should be “effective connectivity” in various places.

Thank you, corrected!

10. The information given in Table 2 does not benefit from a table format.

Although we appreciate that most papers don’t present these details in a table, we believe that it is much better to list them in a way that details of interest can be rapidly found by readers. We have added an additional section to the table to clarify.

11. Methods: The number of items of each type (words and pseudowords) are only presented per run, but it is not clarified how many runs for each stimuli type were included.

Thank you for spotting this. We have added it to the Methods text and Table 2. There was one run for words and one run for pseudowords.

“The current study focused on brain activation elicited when our participants were repeating heard words or pseudowords in different scanning runs (one run for word repetition and one run for pseudoword repetition).”

12. Fig 5a – the lower box is confusing as it does not represent the models used in the current study.

Thank you: we have simplified the neuronal model and removed the modulatory effects. Additionally, we have replaced the EEG image to fMRI formulation.

13. Line 282 – ‘underwriting’ should be ‘underlying’.

Thank you. We have corrected this.

REVIEWERS' COMMENTS:

Reviewer #1 (Remarks to the Author):

I recommend the revised manuscript for publication. The authors did a good job in revising their manuscript.

Reviewer #2 (Remarks to the Author):

The authors have addressed all my concerns.